

# DNA damage-dependent mechanisms of ionizing radiation-induced cellular senescence

Jiebing Guan[1], Tuo Li[1], Feifei Ma[1], Ning Wang[1], Huanteng Zhang[2], Jiale Li[2], Jianguo Li[1], Chang Xu[2] and Qiang Liu[2]

[1] School of Basic Medicine, Shandong Second Medical University, Weifang, China
[2] Institute of Radiation Medicine, Chinese Academy of Medical Sciences & Peking Union Medical College, Tianjin, China

## ABSTRACT

Cellular senescence can be broadly categorized into replicative senescence and stress-induced premature senescence. Replicative senescence mainly results from the progressive shortening of telomeres during successive cell divisions, eventually leading to the arrest of cell division and the onset of senescence. In contrast, stress-induced premature senescence is typically triggered by environmental factors, such as ionizing radiation (IR). While the DNA damage induced by IR has been extensively studied, the specific mechanisms by which IR induces cellular senescence *via* DNA damage remain incompletely understood. This review focuses on IR-induced cellular senescence, particularly in the context of DNA damage. Understanding these mechanisms provides insight into the long-term effects of radiation on cellular senescence and lays the groundwork for future research into the effects of radiation on aging processes.

Corresponding authors
Chang Xu, xuchang@irm-cams.ac.cn
Qiang Liu, liuqiang@irm-cams.ac.cn

## INTRODUCTION

Aging represents a complex biological phenomenon fundamentally characterized by cellular senescence (*Muller, 2009*), driven by intricate molecular mechanisms. Cellular senescence refers to a stable, non-proliferative state marked by cell cycle arrest, diminished cell function, resistance to cell death, upregulated expression of the cell cycle inhibitory gene p16$^{INK4a}$, as well as a series of biological changes within the cellular environment. Cellular senescence manifests in two main forms. The first is replicative senescence (RS), also known as the Hayflick limit, which is caused by progressive telomere shortening with each cell division. When critical telomere length is reached, cells undergo permanent cell cycle arrest (*Hayflick & Moorhead, 1961*). The second type is stress-induced premature senescence (SIPS), which is triggered by external stressors and internal cellular damage, such as oxidative stress, DNA damage, inflammatory responses, ionizing radiation (IR), or exposure to certain chemical toxins. Leonard Hayflick first described the phenomenon of cellular senescence, observing that normal human embryonic cells divide approximately 40–60 times before entering senescence. This phenomenon, known as the 'Hayflick limit',

suggests that the natural human lifespan is capped at around 120 years (*Hayflick, 1965*). This discovery established the foundational concept of cellular senescence and provided a critical experimental model for subsequent aging research. Cellular senescence can also be triggered by both intrinsic and extrinsic factors. Among these, IR is a potent extrinsic inducer of cellular aging, primarily through three interconnected mechanisms: direct induction of DNA damage, activation of cell cycle checkpoint, and amplification of oxidative stress. Together, these insults compromise cellular homeostasis and drive the transition to a senescent state, a process termed SIPS that phenotypically resembles RS (*Suzuki & Boothman, 2008*). At the molecular level, IR inflicts direct genomic damage in the form of base damage, single- and double-strand DNA breaks, which are repaired by pathways such as base excision repair (BER), nucleotide excision repair (NER), homologous recombination (HR), and non-homologous end joining (NHEJ). The cell cycle is tightly regulated by checkpoints (G1/S, intra-S, G2/M) controlled by p53-p21 signaling, which pause the cell cycle to allow repair. While low-doses of radiation typically allow for efficient DNA repair through endogenous mechanisms, higher doses often generate complex, persistent lesions that can evade repair for extended periods. The accumulation of such irreparable DNA damage serves as a critical trigger for senescence initiation (*Rodier et al., 2011*). Therefore, this review focuses specifically on the DNA damage-dependent pathways underlying IR-induced cellular senescence, particularly in scenarios where DNA repair capacity is overwhelmed or compromised.

The target audience of this article primarily encompasses cell biologists with a keen interest in the mechanisms of cellular senescence, DNA damage response, and cell cycle regulation, as well as molecular biologists who are engaged in the study of gene expression, signaling pathways, and protein interactions. Additionally, students and educators in the relevant fields may also use this article as an educational resource to learn about the latest research into IR-induced cellular senescence.

## SURVEY METHODOLOGY

The PubMed database was utilized to search for relevant literature using keywords such as "Ionizing Radiation," "Cellular Senescence," "Senescence," "Aging," "DNA Damage," "Cancer," "DDR," "NHEJ," "ROS," "SASP," "p53," "SA-β-galactosidase," and "p21." Additionally, some of the data utilized in this article were also sourced from Web of Science and Google Scholar. The literature citations aim to provide comprehensive and impartial coverage of the topic. The search criteria encompass a time span of the past 5 to 10 years, and even extend back to the seminal research articles.

## CHARACTERISTICS OF CELLULAR SENESCENCE

The senescent phenotype can be distinguished morphologically by cellular hypertrophy (increased size and flattening), nuclear abnormalities (enlargement or multinucleation), cytoplasmic alterations (enhanced granularity and vacuolization), and abnormal organelle morphology, particularly lysosomal expansion. Senescent cells exhibit four key hallmarks:

(1) permanent cell-cycle arrest mediated by p53-p21 and p16-Rb pathways, (2) a pro-inflammatory senescence-associated secretory phenotype (SASP) that remodels the tissue microenvironment, (3) accumulation of macromolecular damage (*e.g.*, DNA breaks, oxidized proteins, and lipofuscin), and (4) altered metabolism, including mitochondrial dysfunction and enhanced lysosomal activity (*Gorgoulis et al., 2019*; *Humphreys, ElGhazaly & Frisan, 2020*).

The irreversible cell-cycle arrest in senescence is primarily mediated by two interconnected tumor suppressor pathways. The p53-p21 axis responds to DNA damage and oxidative stress, where stabilized p53 transcriptionally activates the cyclin-dependent kinase (CDK) inhibitor p21, blocking cyclin-CDK complexes to prevent Rb phosphorylation and E2F-mediated S-phase entry. In parallel, the p16-Rb pathway reinforces arrest through p16-mediated inhibition of CDK4/6, maintaining Rb in its active, hypophosphorylated state to permanently suppress E2F target genes. This arrest is stabilized by epigenetic silencing and sustained by SASP factors, creating a failsafe mechanism against proliferation. Together, these pathways ensure senescence irreversibility through layered transcriptional and chromatin-level repression of cell-cycle progression (*Sharpless & Sherr, 2015*).

The SASP is a complex and dynamic collection of secreted factors produced by senescent cells. This includes a wide range of molecules, such as pro-inflammatory cytokines (*e.g.*, IL-1α, IL-1β, IL-6, IL-8), chemokines (*e.g.*, CCL2, CXCL1, CXCL10), growth factors (*e.g.*, TGF-β, HGF), matrix metalloproteinases (*e.g.*, MMP1, MMP3), lipids (*e.g.*, prostaglandins), and extracellular vesicles (*e.g.*, exosomes containing microRNAs). In a study by *Jeon et al. (2023)*, it was observed that physiological aging is transferred from old mice to young mice following blood exchange, and it was pointed out that some factors are due to SASP-induced secondary senescence. The SASP is regulated by various signaling pathways, including the DNA damage response (DDR), NF-κB, p38 MAPK, mTOR, and cGAS–STING pathways, as well as epigenetic changes. Its composition and strength vary depending on the duration of senescence, the origin of the pro-senescence stimulus, and the cell type. The SASP plays a crucial role in mediating the non-cell-autonomous effects of senescent cells, influencing tissue microenvironments, immune responses, and contributing to both beneficial and detrimental biological functions in different physiological and pathological contexts (*Wang et al., 2024a*).

Senescent cells exhibit widespread macromolecular damage, encompassing a variety of molecular alterations. A defining feature is DNA damage, including persistent lesions such as double-strand breaks (DSBs), oxidative modifications, and telomere dysfunction. These lesions activate the DDR, triggering cell cycle arrest *via* key regulators such as ataxia-telangiectasia mutated (ATM), ataxia-telangiectasia and Rad3 related (ATR), and p53. Proteins in senescent cells also frequently undergo oxidative modifications—such as carbonylation and nitration—that impair their structure and function. Similarly, lipid peroxidation generates reactive aldehydes like 4-hydroxy-2-nonenal (4-HNE), which further modify proteins and lipids, exacerbating cellular dysfunction. A prominent consequence of this oxidative damage is the accumulation of lipofuscin, an autofluorescent, undegradable pigment that builds up in lysosomes. Composed of cross-linked peroxidation

byproducts (*e.g.*, malondialdehyde and 4-HNE bound to proteins), lipofuscin forms insoluble aggregates that resist lysosomal degradation. This reflects a decline in cellular clearance mechanisms and serves as a key biomarker of senescence and aging (*Nousis, Kanavaros & Barbouti, 2023*).

Cellular senescence is accompanied by substantial metabolic shifts, primarily manifested through changes in mitochondrial and lysosomal functions. The mitochondria of senescent cells often show decreased membrane potential, increased production of reactive oxygen species (ROS), and reduced efficiency in ATP synthesis. It has been found that the pluripotency transcription factor Nanog homeobox (NANOG) can rejuvenate dysfunctional mitochondria in senescent cells. Proline supplementation has been shown to rescue mitochondrial respiratory dysfunction in senescent mesenchymal stem cells by activating mitophagy and restoring metabolic homeostasis, thereby reducing senescence markers (*Choudhury et al., 2024*). Enhanced lysosomal function in senescent cells is marked by elevated levels of lysosomal enzymes, greater lysosomal mass, and enhanced autophagic flux. These factors contribute to the accumulation of undegraded material, such as lipofuscin, and the secretion of SASP. These lysosomal changes, including the elevated activity of the enzyme senescence-associated β-galactosidase (SA-β-gal) and the accumulation of oxidized material, have been particularly exploited as senescence biomarkers (*Debacq-Chainiaux et al., 2009*).

The presence or absence of senescent cells can have harmful effects depending on the context. For example, their accumulation in aged tissues exacerbates chronic inflammation and fibrosis, while their removal in developing organs (*e.g.*, lungs) or during regeneration (*e.g.*, liver) disrupts normal function. In addition, senescent cells were identified in resected glioblastoma multiforme (GBM) tissues from patients and in mouse GBM models, discovering that partially removing p16[INK4a]-positive malignant senescent cells modulates the tumor microenvironment and improves survival outcomes in GBM mice (*Salam et al., 2023*). The definition of senescence remains controversial due to the lack of universal biomarkers—while p16 and SASP are commonly used, their expression varies across cell types. Additionally, senescence exhibits dual roles, acting beneficially in tumor suppression and wound healing but detrimentally in chronic diseases. This functional ambiguity, combined with inconsistent results from animal models, challenges the classification of senescence as purely pathological or physiological. Ultimately, the dynamic nature of senescent cells complicates their therapeutic targeting and necessitates context-specific definitions (*López-Otín et al., 2023*; *De Magalhães, 2024*).

## DNA DAMAGE IS AN IMPORTANT INDICATOR OF CELLULAR SENESCENCE

Cellular senescence is a state of irreversible cell cycle arrest that occurs in response to various stressors, including DNA damage. DNA damage arises constantly due to endogenous and exogenous factors. Under physiological conditions, endogenous instability of DNA leads to spontaneous lesions, such as the hydrolytic cleavage of glycosidic bonds, the deamination of bases and the formation of oxidative adducts. It is estimated that there are 10,000 to
100,000 lesions per cell per day (*Yousefzadeh et al., 2021*). Exogenous sources, including IR and environmental mutagens, further exacerbate this damage. To counteract these threats, cells rely on highly conserved DNA repair pathways. Mammalian cells employ four key repair mechanisms: BER for oxidative lesions, NER for bulky adducts, mismatch repair (MMR) for replication errors, and DSB repair *via* error-prone NHEJ, which is mediated by Ku70/80, or precise HR, which requires the MRE11-RAD50-NBS1 (MRN) complex (*Huang & Zhou, 2021*).

The accumulation of DNA damage, particularly complex and clustered lesions, is a critical driver of senescence, acting as both a cause and a consequence of senescence (*Pustovalova et al., 2025*). The reviewed literature highlights the pivotal role of DNA damage in inducing senescence, especially when the damage is severe or poorly repaired, as is often the case with high-linear energy transfer (LET) radiation (*Mavragani et al., 2019*). Unlike isolated lesions, clustered DNA damage—such as closely spaced DSBs, single-strand breaks (SSBs), and base lesions, poses a great challenge to cellular repair system. When cells fail to adequately resolve such damage, persistent DNA lesions activate the DDR pathway, leading to cell cycle arrest and senescence. For instance, studies have shown that high-LET radiation, such as carbon ions or alpha particles, generates highly clustered DNA lesions that are repaired more slowly or remain unrepaired, resulting in prolonged DDR activation and eventual senescence (*Van De Kamp et al., 2021*). This is supported by the increased expression of senescence markers, such as β-galactosidase, and the formation of senescence-associated heterochromatin foci (SAHF) in irradiated cells (*Helm et al., 2016*).

Evidence from progeroid syndromes, such as Werner syndrome and Hutchinson-Gilford progeria syndrome, highlights the consequences of defective DNA repair and accelerated senescence. These conditions are caused by mutations in genes involved in DNA maintenance (*e.g.*, WRN and LMNA), resulting in patients exhibiting age-related pathologies at an early age (*Kudlow, Kennedy & Monnat, 2007*) . Similarly, studies in mouse models with DNA repair deficiencies, such as Ercc1 mutants, demonstrate increased DNA damage, senescence, and aging phenotypes that mirror natural aging processes (*Vander Linden et al., 2024*). These findings reinforce the idea that DNA damage is a critical indicator of cellular senescence and organismal aging.

## EFFECTS OF IR-INDUCED DNA DAMAGE ON CELLULAR SENESCENCE

IR refers to radiation with sufficient energy to release electrons from atoms or molecules, causing them to become electrically charged. Such radiation can be electromagnetic waves or particle streams, including alpha rays, beta rays, X-rays, gamma rays, and neutron radiation (*Fiorentino et al., 2009*). IR causes cell senescence mainly by inducing DNA damage, halting cell cycle progression, increasing SASP secretion, increasing oxidative stress, and shortening telomeres (Fig. 1). However, *Soroko et al. (2024)* found that low-dose-rate (LDR) radiation significantly induces cellular senescence, manifested as an increase in SA-β-gal stain-positive cells, and this effect was dose-dependent. High-dose-rate (HDR) radiation did not significantly induce cellular senescence, which may be related

to the increased apoptosis it causes. *Aboussekhra et al. (2023)* found that different doses of IR inhibit the activity of breast cancer-associated fibroblasts and their adjacent fibroblasts. Importantly, β-gal staining of breast stromal fibroblast cells was significantly increased after high-dose X-ray (50 Gy) treatment, demonstrating that high doses of IR induce senescence. IR can induce severe DNA damage, often leading to apoptosis and cellular senescence *in vitro* and *in vivo*, which is associated with delayed repair of irradiated tissues (*Wang et al., 2019*). IR induces cellular senescence and significantly impacts the expression of the SASP. Chronic DNA damage is also known to induce a high secretory state of SASP (*Wang et al., 2024a*). For example, in a study of human esophageal squamous cell carcinoma, it was found that radiotherapy can induce senescence in cancer cells and stimulate nearby cells through paracrine secretion of pro-inflammatory factors (*Zhang et al., 2023*). In studies of glioblastoma, *Jeon et al. (2023)* discovered that IR-induced tissue factor (F3) was strongly induced in senescent glioblastoma (GBM) cells, and F3 coordinates the remodeling of the oncogenic tumor microenvironment (TME) by activating tumor-autonomous signaling and the extrinsic coagulation pathway. In an investigation of the effects of IR on the induction of senescence in annular fibroblast (AF) cells, AF cells were exposed to 10-15 Gy of IR for five days. This exposure resulted in near-complete senescence, as evidenced by increased SA-β-gal staining and a significant elevation in SASP expression (*Zhong et al., 2022*). *Isermann, Mann & Rübe (2020)* found that the histone variant H2A.J is closely associated with IR-induced cellular senescence and the DDR. In H2A.J-deficient mice, there was a notable increase in SASP factor secretion in the skin. H2A.J regulates the expression of SASP-related genes through its influence on chromatin structure, thereby playing a critical role in modulating the SASP during IR-induced senescence. Certain SASP factors accelerate cell senescence, with IL-6 and IL-8 being the most extensively studied. IL-6 and IL-8, two pro-inflammatory cytokines, are typically expressed at high levels in senescent cells. Elevated IL-6 and IL-8 levels play a crucial role in reinforcing senescence and promoting inflammation (*Wang et al., 2024b*).

IR generates a substantial amount of ROS, which can cause oxidative damage to cellular macromolecules including DNA, proteins, and lipids (*Terman, 2001*). Moreover, increased production of ROS within the cell is closely associated with the accumulation of dysfunctional mitochondria (*Guilbaud, Sarosiek & Galluzzi, 2024*). Radiation induces cellular senescence and the accumulation of dysfunctional mitochondria in salivary gland organoids, leading to disrupted mitochondrial dynamics, reduced mitochondrial DNA copy number, and increased ROS production (*Cinat et al., 2024*). Research indicates that nicotinamide riboside (NR) can significantly alleviate intestinal aging induced by IR, with mechanisms including reducing oxidative damage, restoring normal function of intestinal stem cells, regulating disruption of the gut symbiotic ecosystem, and resolving metabolic abnormalities (*Yue et al., 2024*). In addition, *Zhong et al. (2022)* demonstrated that IR induces cellular senescence in annulus fibrosus cells through the oxidative stress pathway and activates MMP-mediated matrix degradation pathways, leading to degeneration and dysfunction of the intervertebral disc tissue. Therefore, an increase in ROS or a decrease in antioxidant responses will accelerate cellular senescence.

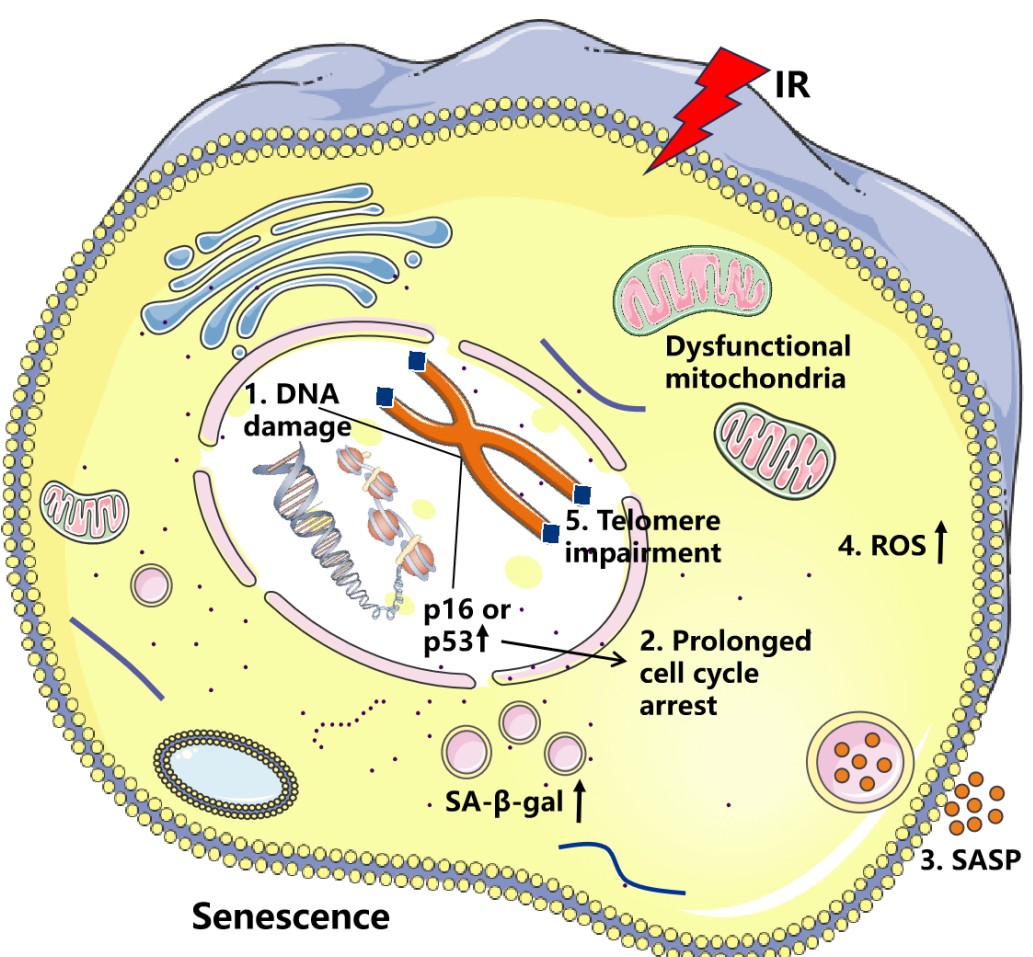

**Figure 1 Effects of ionizing radiation (IR) on cellular senescence.** IR induces DNA damage and elevates reactive oxygen species (ROS) levels, resulting in increased p16 and p53 activity, which promotes cell cycle arrest. Sustained DNA damage signaling enhances both the secretion of senescence-associated secretory phenotype (SASP) factors and ROS production. This creates a vicious cycle of oxidative DNA damage and telomere attrition, ultimately culminating in cellular senescence.

Here, we propose that radiation-induced DNA damage triggers cellular senescence through three distinct yet interconnected mechanisms. First, radiation-mediated DNA damage can induce cellular growth arrest by activating cell cycle checkpoints (*Campisi, 2013*). This protective response allows cells to temporarily halt proliferation upon detecting DNA lesions, providing time for DNA repair while preventing the transmission of damaged genetic material during replication (*Bekker-Jensen & Mailand, 2010*). Second, IR generates severe DNA lesions, including DSBs or DNA crosslinks, which frequently exceed the cell's intrinsic repair capacity (*Rodier et al., 2009*). The accumulation of unrepaired DNA damage leads to persistent genomic instability, ultimately compromising cellular viability. Third, IR can inflict damage to telomeric DNA, thereby accelerating the senescence process. Table 1 summarizes the key evidence from recent studies that supports the three proposed mechanisms by which radiation-induced DNA damage triggers cellular senescence. Cellular

**Table 1  Effects of radiation-induced cellular senescence.**

| Model | Radiation source | Dose | Signaling | Effects | Reference |
|---|---|---|---|---|---|
| MCF7 cells, VH10tert cells, and RPE-1 cells | γ-rays | 5 Gy | p21-CDK4-DREAM | DREAM complex suppresses cell cycle-related genes, promoting senescence | *Schmidt et al. (2024)* |
| Human fetal lung and skin fibroblasts | γ-rays X-rays | 1 Gy 0.6 Gy | γ-H2AX | SAHF formation, promoting senescence | *Oizumi et al. (2024)* |
| Head and neck squamous cell carcinoma (HNSCC) cell lines | X-rays | 2 Gy and 4 Gy | p53-p21 | Inhibits DNA damage repair, enhances IR-induced cellular senescence | *Dobler et al. (2020)* |
| Human aortic endothelial cells (HAECs) | X-rays | 3 Gy, 5 Gy, and 10 Gy | BRCA1-BARD1-RAD51 | Enhances DNA damage repair | *Park et al. (2022)* |
| Breast cancer and melanoma cell models | X-rays | 6 Gy and 15 Gy | BRCA1-RAD51 | Delays DNA repair and promotes cellular senescence | *Efimova et al. (2018)* |
| Fibroblasts | X-rays | 1 Gy, 2 Gy, 5 Gy and 10 Gy | p53-p21 | Inhibits DNA damage repair | *Osipov et al. (2023)* |
| Mouse alveolar type II epithelial stem cells (AEC2) | X-rays | 5 Gy and 8 Gy | FBW7-TPP1 | Triggers telomere uncapping, leading to cellular senescence | *Wang et al. (2020)* |
| Vascular smooth muscle cells (VSMCs) | X-rays | 4 Gy and 8 Gy | NF-κB-CTCF-p16 | Cell cycle arrest, cells enter senescence | *Zheng et al. (2024)* |

senescence represents the terminal outcome of these cascading events. Prolonged exposure to DNA damage and diminished repair capacity eventually leads to cellular functional decline and the onset of aging.

## Radiation-induced DNA damage triggers cell cycle arrest leading to cellular senescence

The cell cycle checkpoint pathway is a critical biochemical surveillance system which halts cell cycle progression upon DNA damage detection (*Nyberg et al., 2002*). When IR induces DNA damage, this system principally operates through three main cell cycle checkpoints—the G1/S, intra-S and G2/M checkpoints—to arrest the cell cycle and initiate DNA damage repair (*Sancar et al., 2004*). Each checkpoint is governed by specific proteins that sense the damage and initiate signaling cascades. In the cell cycle, the activation of cyclin-dependent kinase 2 (CDK2) is crucial for regulating both the G1/S transition and the subsequent S-phase progression (*Zannini, Delia & Buscemi, 2014*). Proper cell cycle advancement requires precise regulation of CDK2 phosphorylation status—its dephosphorylation and activation are essential for S-phase entry (*Falck et al., 2001*). The phosphatase Cell Division Cycle 25A (CDC25A) plays a pivotal role in this process by specifically removing inhibitory phosphates from CDK2, thereby enabling cell cycle progression. However, phosphorylation of CDC25A triggers its functional inactivation, followed by ubiquitin-mediated degradation (*Matsuoka, Huang & Elledge, 1998*). The degradation of CDC25A hinders the dephosphorylation of CDK2, preventing cells from entering the S phase and causing cell cycle arrest (*Falck et al., 2001*).

Upon detection of radiation-induced DNA damage, including DSBs and base lesions, cells initiate a sophisticated damage response through the activation of master checkpoint

kinases ATM and ATR (*Maréchal & Zou, 2013*). These phosphoinositide 3-kinase-related kinases (PIKKs) transmit signals by phosphorylating their downstream effector kinases, checkpoint kinase 1 (CHK1) and checkpoint kinase 2 (CHK2). CHK1 is a key cell cycle checkpoint protein, and its activity is normally regulated by the phosphorylation of ATR protein kinases (*Ciccia & Elledge, 2010*). ATR is a signal transduction protein kinase that detects SSBs in cells. When cells sense DNA damage, CHK1 is activated by ATR phosphorylation, allowing it to localize to the damage site. Phosphorylated CHK1 plays a crucial role in the G1/S and G2/M transitions of the cell cycle (*Ciccia & Elledge, 2010*; *Maréchal & Zou, 2013*). CHK1 prevents activation of the CDK complex by phosphorylating CDC25A, thereby inhibiting its phosphatase activity and leading to cell cycle arrest and the induction of cellular senescence. CHK2 is another checkpoint protein that functions during the G1/S and G2/M transition phases of the cell cycle when DNA damage is detected. ATM activates CHK2 by directly phosphorylating it, and activated CHK2 phosphorylates Rb to promote G1/S arrest. Additionally, CHK2 phosphorylates p53 to increase p21 levels, thereby maintaining G2/M arrest (*Zannini, Delia & Buscemi, 2014*).

Radiation-induced DNA damage typically results in cellular senescence *via* two tumor suppressor pathways: the p53—p21$^{CIP1}$ axis and p16$^{INK4a}$—Rb pathway (*Mijit et al., 2020*). p21$^{CIP1}$ plays dual roles by initially inducing transient G2/M arrest through CDK1/2 inhibition, followed by permanent growth arrest *via* the p21$^{CIP1}$/CDK4/DREAM complex that irreversibly blocks cell cycle progression (*Schmidt et al., 2024*). Studies have found that IR can significantly induce the aging of brain cells (such as neurons and glial cells), manifested by the fact that p53 is up-regulated by p21$^{CIP1}$, and that p16$^{INK4a}$ is jointly responsible for cell cycle arrest and increased senescence-related β-galactosidase activity by inhibiting Rb protein phosphorylation (*Wang et al., 2021*; *Zhong et al., 2024*). Furthermore, IR (especially high-dose radiation) causes DSBs in vascular endothelial cells and activates the ATM/ATR-Chk1/Chk2-p53-p21$^{CIP1}$ signaling pathway, leading to cell cycle arrest and aging (*Nagane et al., 2021*). Recently, it has been found that the NF-κB/CTCF/p16$^{INK4a}$ pathway plays a crucial role in the senescence of vascular smooth muscle cells (VSMCs) induced by IR. Specifically, IR activates the NF-κB pathway, promoting the nuclear translocation and phosphorylation of the p65 subunit, thereby enhancing its DNA-binding capacity. The activation of NF-κB also regulates the expression and spatial clustering of CCCTC-binding factor (CTCF), which interacts with the p16 gene to promote p16 expression. Furthermore, downregulation of High Mobility Group Box 2 (HMGB2) expression is closely related to CTCF's spatial clustering, and the reduction of HMGB2 further enhances CTCF's regulation of the p16 gene. Ultimately, the upregulation of p16$^{INK4a}$ and p21$^{CIP1}$ leads to cell cycle arrest, causing the cells to enter a senescent state (*Zheng et al., 2024*). Thus, cell cycle regulatory proteins are essential for maintaining normal cell physiology and play a critical role in regulating the onset and progression of cellular senescence (Table 2).

## Radiation-induced DNA damage exceeds the repair capacity of cells, leading to cellular senescence

IR is a potent genotoxic agent that induces a range of DNA lesions, ranging from simple SSBs to highly complex clustered DNA damage. The severity of these lesions depends

**Table 2  DNA damage-induced cell cycle arrest and senescence.**

| Key Molecule/Pathway | Function & Regulatory Mechanism | Downstream effect | References |
|---|---|---|---|
| CDK2 | Drives G1/S transition; inhibited by p21$^{CIP1}$ and CDC25A loss | G1/S arrest | *Zannini, Delia & Buscemi (2014)* and *Falck et al. (2001)* |
| CDC25A | Dephosphorylates/activates CDK2; degraded after CHK1 phosphorylation | Inhibits S-phase entry | *Matsuoka, Huang & Elledge (1998)* and *Maréchal & Zou (2013)* |
| CHK1 | Phosphorylated by ATR; prevents activation of the CDK complex by phosphorylating CDC25A | G1/S and G2/M arrest | *Ciccia & Elledge (2010)* and *Maréchal & Zou (2013)* |
| CHK2 | Phosphorylated by ATM → activates p53 and Rb | G1/S and G2/M arrest | *Zannini, Delia & Buscemi (2014)* |
| p53-p21$^{CIP1}$ pathway | CHK2 phosphorylates p53 → upregulates p21$^{CIP1}$ → inhibits CDK1/2/4 | G2/M arrest | *Schmidt et al. (2024)*, *Wang et al. (2021)*, *Zhong et al. (2024)* and *Nagane et al. (2021)* |
| NF-κB/CTCF/p16$^{INK4a}$ pathway | Radiation activates NF-κB → CTCF clustering → activates p16$^{INK4a}$ and inhibits HMGB2 | S and G2/M arrest | *Zheng et al. (2024)* |

on the radiation quality, including LET, particle type, and energy. High-LET radiation, such as alpha particles and carbon ions, is particularly effective at generating clustered DNA damage, which consists of multiple lesions, including DSBs, SSBs, and oxidized bases, within a localized region of the DNA (*Hada & Georgakilas, 2008*). These complex lesions pose a significant challenge to cellular repair mechanisms, often exceeding the cell's capacity for efficient repair. When the damage is not adequately repaired, it can trigger cellular senescence that serves as a protective mechanism against genomic instability and carcinogenesis but also contributes to tissue aging and dysfunction. Clustered DNA damage is a hallmark of high-LET radiation (*Sage & Shikazono, 2017*). Unlike isolated lesions, which are typically repaired by BER or NER, clustered lesions require coordinated action from multiple repair pathways. For example, a DSB flanked by oxidized bases or additional SSBs may stall the repair machinery, as the presence of nearby lesions interferes with the recognition and processing of the primary break. The repair of such lesions often results in incomplete or erroneous repairs, contributing to genomic instability. A key consequence of complex DNA damage is the delayed or incomplete activation of repair pathways. Studies have shown that DSBs induced by high-LET radiation are repaired more slowly than those caused by low-LET radiation. For instance, phosphorylated histone H2AX (γ-H2AX) foci, markers of DSBs, persist longer in cells exposed to high-LET radiation, indicating unresolved damage (*Antonelli et al., 2015*). This delay is attributed to the requirement for additional repair factors and the difficulty in processing lesions within condensed chromatin regions. Moreover, the repair of non-DSB clustered lesions, such as oxidized bases near SSBs, can inadvertently generate secondary DSBs during BER, further exacerbating the damage burden. Over time, the accumulation of unresolved lesions can overwhelm the cell's repair capacity, leading to the activation of senescence pathways.

On the other hand, a decline in repair capacity also contributes to the onset of cellular senescence. Defects in DNA repair mechanisms, as observed in premature aging disorders such as Werner syndrome (WS), xeroderma pigmentosum (XP) and Cockayne syndrome (CS), underscore the vital function of DNA repair in maintaining genomic stability. Cells from WS patients display pronounced genomic instability, including increased chromosomal aberrations and deletions, due to WRN dysfunction impairing DNA repair pathways and exacerbating cellular senescence and aging-related disorders (*Lu & Davis, 2021*). XP patients exhibit extreme sensitivity to UV radiation, resulting in premature skin aging and a high risk of skin cancer (*Rizza et al., 2021*). In contrast, CS is characterized by systemic premature aging, demonstrating that impaired BER and NER can contribute to accelerated aging phenotypes (*Marteijn et al., 2014*). These aging-related diseases are closely associated with defects in DNA damage repair. BRCA1 Associated RING Domain 1 (BARD1) is an important protein involved in DNA damage repair, and its dysfunction has been linked to cellular senescence. *Park et al. (2022)* showed that IR downregulates BARD1 in human aortic endothelial cells (HAECs), leading to γ-H2AX accumulation and increased SA-β-gal activity, thereby inducing the senescence of HAECs. This finding underscores how radiation-induced suppression of repair mechanisms accelerates cellular aging. Other studies indicate that the expression levels of BRCA1 and Rad51, two critical HR repair proteins, significantly decrease under conditions of persistent DNA damage (*Efimova et al., 2018*). The absence of these proteins compromises the ability of cells to perform HR repair efficiently, which may lead to DNA damage accumulation, potentially triggering cell dysfunction, senescence, or even cell death. This mechanism could be exploited in cancer treatment to induce senescence in tumor cells by leveraging their deficient DNA repair ability, thereby inhibiting their proliferation and spread. The interplay between radiation and DDR pathways is crucial in determining the fate of cells. Combining IR with DDR inhibitors (*e.g.*, ATM/ATR inhibitors) has been shown to significantly increase senescence in head and neck squamous cell carcinoma cells, suggesting that suppressing repair pathways can exacerbate radiation-induced senescence (*Dobler et al., 2020*). Similarly, it has been demonstrated that 3-Hydroxy-3-Methylglutaryl Coenzyme A (HMG-CoA) reductase inhibitors can enhance the persistence of DSBs, inducing senescence in tumor cells (*Vermeij, Hoeijmakers & Pothof, 2016*). DDR proteins form lesions when cells respond to DNA damage, and these lesions can still be observed 24 h after irradiation and later, which are called "residual lesions". They are considered to be repair sites for complex and potentially fatal DSBs. This proves that residual DDR lesions play an important role in cell aging and helps to further explore the influence of radiation on cell fate. However, *Osipov et al. (2023)* discovered in their research on the relationship between the residual lesions caused by IR and the aging of cells that the number of residual lesions in fibroblasts gradually decreased with the extension of irradiation time, while the proportion of cell aging gradually increased. A study by *Oizumi et al. (2024)* revealed that SAHF physically obstructs the phosphorylation of H2AX at DSB sites, delaying early DDR signaling. This suppression was particularly pronounced in senescent cells, where SAHF formation is elevated due to chromatin reorganization during aging. These findings underscore SAHF as a key contributor to the decline in DSB repair efficiency during radiation-induced cellular

**Table 3   DNA repair deficiencies and radiation-induced senescence.**

| Key Molecule/Disease | Function & Findings | Phenotype/Effect | References |
|---|---|---|---|
| Werner syndrome (WS) | HR/NHEJ defects | Genomic instability, chromosomal aberrations and deletions | *Lu & Davis (2021)* |
| Xeroderma pigmentosum (XP) | NER defects | Photosensitivity, cancer predisposition | *Lu & Davis (2021)* and *Rizza et al. (2021)* |
| Cockayne syndrome (CS) | NER/BER defects | Multisystem progeria, neurodegeneration | *Marteijn et al. (2014)* |
| BARD1 | Increased γ-H2AX foci formation; elevated SA-β-gal activity | HAEC senescence | *Park et al. (2022)* |
| BRCA1/Rad51 | HR defects | Tumor suppression and senescence | *Efimova et al. (2018)* |
| DDR inhibitors (DDRi) | ATM/ATR inhibitors and radiation | HNSCC senescence | *Dobler et al. (2020)* |
| HMG-CoA reductase inhibitors | Prolonged persistence of DSBs | Tumor cell senescence | *Vermeij, Hoeijmakers & Pothof (2016)* |
| SAHF | Suppress the formation of γ-H2AX foci; heterochromatin condensation | Delay DNA repair | *Oizumi et al. (2024)* |

senescence. Table 3 outlines the mechanism of DNA damage accumulation resulting from deficiencies in DNA repair.

## Ionizing radiation induces telomere DNA damage and cellular senescence

Telomeres are specialized nucleoprotein structures located at the ends of eukaryotic chromosomes, consisting of repetitive TTAGGG DNA sequences and associated shelterin proteins. These protective caps play a crucial role in maintaining genomic stability by preventing chromosome end-to-end fusions and shielding chromosomal termini from being recognized as DSBs. However, telomeres progressively shorten with each cell division due to the end-replication problem, whereby DNA polymerase fails to fully replicate the lagging strand. When telomeres become critically short, they lose their protective function, triggering DDR and leading to RS (*Di Micco et al., 2021*).

IR exacerbates telomere dysfunction by directly inducing DNA damage, including DSBs and oxidative lesions within telomeric regions. Radiation-induced telomere damage resembles natural telomere shortening and recruits key DDR proteins, such as 53BP1, γ-H2AX, and the MRN complex, to telomeric sites. The activation of ATM kinase at damaged telomeres initiates a signaling cascade resulting in cell cycle arrest, primarily through the p53/p21$^{CIP1}$ and p16$^{INK4A}$/Rb pathways. These molecular events not only halt proliferation but also drive cells into a senescent state characterized by morphological changes, increased SA-β-gal activity and the secretion of SASP.

The structural and functional integrity of telomeres is maintained by the shelterin complex, a group of specialized proteins including TRF1, TRF2, and POT1. TRF1 and TRF2 bind directly to double-stranded telomeric DNA, while POT1 associates with single-stranded overhangs, collectively preventing inappropriate DDR activation. For example, experimental inhibition of TRF2 leads to rapid telomere deprotection, resulting in ATM-dependent DDR signaling and p53/pRb-mediated senescence (*De Lange, 2005*). Telomere "uncapping"—the disruption of this protective architecture—activates DDR

pathways similar to those triggered by IR-induced DNA damage, involving both ATM and ATR kinases (*Jiang et al., 2008*; *Maréchal & Zou, 2013*). Notably, ATR exhibits prolonged retention at telomeric lesions following IR exposure, suggesting its critical role in sustaining the senescence phenotype (*Guo et al., 2007*). The DNA replication machinery causes chromosome ends to fail to replicate completely, resulting in telomere shortening with each cell division (*Maréchal & Zou, 2013*).

Emerging research highlights the role of the E3 ubiquitin ligase, F-Box and WD Repeat Domain Containing 7 (FBW7), in regulating telomere stability and senescence. In alveolar epithelial type 2 cells, IR or oxidative stress upregulates FBW7 expression, leading to telomere uncapping and DDR activation. FBW7-mediated telomere dysfunction is associated with increased binding of DNA damage markers ($\gamma$H2AX, 53BP1) to telomeres, elevated levels of cell cycle inhibitors (p53, p21, p16), and G2/M phase arrest (*Wang et al., 2020*). Moreover, FBW7 overexpression accelerates telomere shortening and fragility, promoting SIPS.

The exceptional sensitivity of telomeres to damage stems from several intrinsic characteristics. Their G-rich sequences are particularly susceptible to oxidative stress. Their nuclear membrane tethering may restrict the accessibility of repair factors (*Taddei & Gasser, 2012*). Their condensed heterochromatic structure likely impedes efficient DNA repair processes. These features make telomeric DNA disproportionately vulnerable to genotoxic insults compared to other genomic regions (*Hewitt et al., 2012*).

The consequences of telomere shortening present a paradox in cancer biology. On one hand, telomere attrition acts as a tumor-suppressive mechanism by inducing senescence in pre-malignant cells. On the other hand, excessive shortening can lead to genomic instability, chromosomal rearrangements, and increased cancer risk—particularly in aging populations where telomere reserves are depleted (*Hernández et al., 2015*). In elderly individuals, critically short telomeres fail to protect chromosome ends, promoting oncogenic transformation (*Qiu et al., 2019*). Thus, while IR-induced telomere damage may contribute to radiation-induced senescence, it could also inadvertently fuel carcinogenesis in surviving cells.

## CONCLUSION

IR, a potent genotoxic stressor, induces diverse DNA lesions, including base damage, SSBs, and DSBs, with clustered damage—particularly from high-LET radiation—posing a significant challenge to repair systems. When repair capacity is overwhelmed, persistent DNA damage triggers the DDR, activating cell cycle checkpoints *via* the ATM/ATR-CHK1/CHK2 axis and reinforcing senescence through the p53-p21 and p16-Rb pathways. These cascades lead to irreversible cell cycle arrest, a defining feature of senescence, while simultaneously promoting the SASP secretion. The SASP not only perpetuates senescence but also fosters a pro-inflammatory microenvironment. Additionally, IR-induced telomere dysfunction mimics replicative senescence by recruiting DDR proteins to chromosomal ends. Mitochondrial dysfunction and ROS production further amplify DNA damage, creating a vicious cycle.

Although persistent DSBs typically trigger apoptosis or senescence, the fate of irradiated cells depends on their molecular context. The role of IR-induced senescence in cancer treatment is a double-edged sword. On one hand, IR-induced senescence directly inhibits tumor proliferation by permanently arresting the cell cycle (dependent on the p53-p21 and p16-Rb pathways). Specific factors in SASP (*e.g.*, CXCL10, TRAIL) activate immune surveillance and recruit natural killer cells and cytotoxic T lymphocytes to clear tumor cells (*Prasanna et al., 2021*). On the other hand, senescence escape, observed in around 5% of p53-mutant tumors, enables epigenetic reprogramming (*e.g.*, hypermethylation of the p16 promoter) and re-entry into the cell cycle, often conferring aggressive, therapy-resistant phenotypes (*O'Sullivan et al., 2024*). Therefore, while senescence serves as a defense against tumorigenesis, cancer cells can exploit this process to promote survival.

In summary, IR-induced cellular senescence is primarily driven by DNA damage and its downstream signaling. Understanding these mechanisms not only elucidates the effects of radiation on aging and cancer, but also informs strategies to mitigate radiation risks and harness senescence for therapeutic benefit.

### Funding
This work was funded by the National Natural Science Foundation of China (82273580), the Natural Science Foundation of Shandong Province, China (ZR2022MH011). The funders had no role in study design, data collection and analysis, decision to publish, or preparation of the manuscript.

### Grant Disclosures
The following grant information was disclosed by the authors:
The National Natural Science Foundation of China: 82273580.
The Natural Science Foundation of Shandong Province, China: ZR2022MH011.

### Competing Interests
The authors declare there are no competing interests.

### Author Contributions
- Jiebing Guan conceived and designed the experiments, performed the experiments, analyzed the data, prepared figures and/or tables, and approved the final draft.
- Tuo Li performed the experiments, authored or reviewed drafts of the article, and approved the final draft.
- Feifei Ma performed the experiments, authored or reviewed drafts of the article, and approved the final draft.
- Ning Wang performed the experiments, authored or reviewed drafts of the article, and approved the final draft.
- Huanteng Zhang performed the experiments, authored or reviewed drafts of the article, and approved the final draft.

- Jiale Li performed the experiments, authored or reviewed drafts of the article, and approved the final draft.
- Jianguo Li conceived and designed the experiments, authored or reviewed drafts of the article, and approved the final draft.
- Chang Xu conceived and designed the experiments, analyzed the data, prepared figures and/or tables, authored or reviewed drafts of the article, and approved the final draft.
- Qiang Liu conceived and designed the experiments, authored or reviewed drafts of the article, and approved the final draft.

### Data Availability

This is a literature review.

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
