# Peer review of "DNA damage-dependent mechanisms of ionizing radiation-induced cellular senescence"

_PeerJ, doi:10.7717/peerj.20087_

## Round 0.1 · original submission · Major Revisions

Please address all the concerns raised by the reviewers.

Reviewer 1 ·

Basic reporting

The review entitled “DNA damage-dependent mechanisms of ionising radiation-induced cellular senescence” offers a comprehensive perspective on the effects of ionising radiation on DNA damage and how this is associated with the induction of senescence. The authors use current literature and cover basic concepts of DNA repair, DNA damage, and cell cycle arrest. They illustrate this relationship with clinical examples such as therapeutic agents and diseases. Even though the topic has been recently reviewed (Chen et al., 2019, Oncology Reports; Ibragimova et al., 2024, Cells), the present review offers a more fundamental view of the topic, appealing to a wide audience from basic researchers to clinicians.
The “Introduction” clarifies the target audience and focus of the review. Unfortunately, it is poorly structured and difficult to follow, especially lines 48-52. This section needs a better flow and reorganisation of the ideas. For example, in line 45, the Authors mentioned “two main types”, but only introduced one. The Authors should introduce both types of senescence, which can be better defined later on. Some paragraphs in this section are repeated, for example, lines 57-63.

Experimental design

My main concern is the structure of the review. There is redundancy between the sections entitled “Cellular senescence induced by ionising radiation” and “Effect of DNA damage caused by ionising radiation on cellular senescence.” The latter section is more structured and informative than the former. I suggest merging both sections, transferring some parts from the first to the second. A tentative structure of the review would be:
Characteristics of cellular senescence (this section should also be improved, see comment below)
DNA damage is an important indicator of cellular ageing
Effects of IR-induced DNA damage on cellular senescence.

Furthermore, the Authors are asked to make sure that all the in-text citations are properly formatted as surname’s author + et al.

Validity of the findings

The Introduction should be rewritten to provide the readers with a broader survey of the topic. Even though the goal of the review is properly stated, a reorganisation of the sections and a summary of basic concepts (DNA damage repair pathways and regulation of the cell cycle) would greatly benefit the present review.

The Authors made an interesting point in line 112, arguing that the definition of senescence remains controversial, but they did not discuss why. I think this might be an interesting idea to develop further in the "Conclusion".

Additional comments

Major comments
- Section “Characteristics of cellular senescence”. The hallmarks of cellular senescence should be better explained. The Authors merely mentioned the irreversible growth arrest and the SASP, but molecular details that define these features are lacking. A main feature of senescent cells is their altered metabolism, characterised by the positive β-galactosidase staining, possibly due to the lysosomal activity (Debacq-Chainiaux et al., 2009, Nat Protoc). Morphological changes are also observed upon senescence, which are also missing in this section (move it from lines 52-53). Some literature reviews have nicely covered this topic (Humphreys et al., 2020, Cells; Gorgoulis et al., 2019, Cell).

- Line 95. The concept of SASP should be better explained because, as the Authors discuss, its relevance for the resolution of DNA damage and induction of senescence is critical. I suggest including, at this stage, the specific cytokines/metalloproteins/growth factors that are part of it. Another interesting point to discuss is whether the composition of the SASP changes depending on the trigger (ionising radiation, physiological ageing, other stresses). The definition of SASP (line 157) could be moved to this part of the text.

- Line 110. The Authors mention that the presence or absence of senescent cells in the tissues may be harmful. It would be interesting to give some examples or offer readers some references to consult. The same comment goes for line 111.

- Line 112. “The definition of senescence remains controversial”. The Authors should explain why they consider this, and what the controversy is around it in this context.

- Lines 449-455. This is perceived as a repetition of what the Authors discussed in lines 233-267.

- Figure 1. There is no in-text reference to this Figure.
The figure depicts “DNA damage”, but this does not correlate with the representation.
The Authors may include a neighbouring cell to illustrate the paracrine effect of SASP.
The text box “p16 or p53” should be moved to the nucleus, where the modulation of gene expression occurs. I suggest including another text box or representation entitled “Cell Cycle Arrest”, as this is a hallmark of senescence.
- No references to Table 1 are found in the text.

Minor comments

- Please make sure that the abbreviations are properly presented. For example, if the Authors spell “ionising radiation (IR)” at the beginning of the text, they should use “IR” from that point on throughout the text.

- Please make sure that when spelling out abbreviations, it is always done consistently by first specifying the full name, followed by the abbreviation. For example, ionising radiation (IR).

- Lines 41 and 107. “p16INK4a”: INK4a should be superscript.
- Line 66. “Viability” instead of “vitality”.
- Line 127. SA-β-gal should be properly spelled.
- Line 146. ATM and ATR abbreviations are not properly spelled out.
- Line 172. “Discovered found” duplicated.
- Line 299. No need to spell out ATM again.
- Lines 233-267. I suggest the Authors summarise how DNA damage is resolved, as this is not the subject of the review. A brief summary of this part and a reference to specific literature reviews on the topic would work better.
- Lines 378-401. I suggest the Authors summarise the cell cycle regulation and refer to a specific literature review on the topic.
- Lines 411-412. The sentence “Galactosidase activity increased” seems not to be connected to the rest of the text. I suggest that the Authors rephrase it.
- Line 463. The abbreviation for Cockayne Syndrome seems unnecessary, as it is not used further in the text. The Authors should spell out what MAP means.

·

Basic reporting

This is a review study on a very important study, that of radiation-triggered senescence. The authors have a relatively good overview and refer to some cases relevant to their conclusions.
Unfortunately, the authors miss some important data and findings related to radiation damage and its effects on senescence or cell death.
For example, they need to discuss the fact that ionising radiation induces not only oxidation but primarily clustered DNA damage with quite a few detrimental effects, including genomic instability, cell death, and most likely senescence (see, for example: Cancers (Basel) 2019 Nov 14;11(11):1789 doi: 10.3390/cancers11111789.

In addition, there is a major review that authors need to advise and discuss in relation to their work:
JNCI: Journal of the National Cancer Institute, Volume 115, Issue 2, February 2023, Page 235, https://doi.org/10.1093/jnci/djac225

Overall, this reviewer is very sceptical of the originality and new findings that are brought up in this case, which have not been discussed previously in major studies.

Experimental design

The design of this review study is relatively simple and straightforward, but as currently organised, it does not provide any clear mechanistic link to radiation-induced senescence.

Validity of the findings

The findings are not new, but have been gathered in a relatively acceptable manner.

Reviewer 3 ·

Basic reporting

The manuscript is generally well-written and presents a clear narrative flow; however, there is a notable issue with redundant content in the introduction. Specifically, lines 57–63 (“Cellular senescence can be divided…” through “enter a state similar to aging”) appear to be a near-verbatim repetition of an earlier passage and may have been inserted unintentionally. Removing or revising this repeated content will improve clarity and prevent reader confusion.

Experimental design

The overall organization of the review is logical, with well-defined sections. However, the structure of the section on “Effect of DNA Damage Caused by Ionizing Radiation on Cellular Senescence” could be improved for clarity. The subsections contain dense technical information that would benefit from being accompanied by dedicated summary tables. Currently, there is only one table, and its intended placement and scope are unclear. Adding individual summary tables for each major mechanism discussed in the subsections would enhance readability and help distill key points for the reader.

Validity of the findings

The technical content appears sound, however the placement and referencing of the figure are suboptimal. The figure is not clearly referenced in the text; placing the figure after line 374 would improve interpretability. Additionally, numbering the three primary mechanisms in the figure would strengthen its connection to the corresponding textual discussion and enhance the overall coherence of the manuscript.

Additional comments

Consider providing clearer cross-referencing between figures, tables, and relevant text passages throughout the manuscript.

---

## Round 0.2 · Minor Revisions

Please address the remaining comments.

Reviewer 1 ·

Basic reporting

The Authors have made substantial changes to improve the quality of the review and have addressed most of the comments raised during previously.
The addition of tables to clarify general concepts has helped, however they are not properly introduced in the text. In this regard, I suggest the Authors to re-formulate the cross-reference to all the tables and highlight their content in the text.

Minor comments
- Use the same format when using in-line citations (example, discrepancies in line 201 and line 211).
- Line 199: "high-dose rate...", "High" should be capitalised.

Experimental design

No comment

Validity of the findings

No comment

Additional comments

No comment

·

Basic reporting

The revised version of the manuscript stands much better and it is now more interesting and inclusive.

Experimental design

This is a review study where the essential design has been developed correctly.

Validity of the findings

The statements are valid

---

## Round 0.3 · accepted · Accept

Thanks for addressing all comments!